# OpenReview forum: "Is Safety Standard Same for Everyone? User-Specific Safety Evaluation of Large Language Models"
_NeurIPS.cc/2025/Workshop/Reliable_ML — NeurIPS 2025 - Reliable ML Workshop_

### Official Review · Reviewer_2cxj · 2025-09-17
**Important contribution and good execution**

**Rating:** 8
**Confidence:** 4

**Review:**

# Summary

The authors introduce user-specific safety wherein one needs to ensure that LLMs take into account the user's profile when giving responses; some responses considered benign in general might be harmful if given to specific user groups. The paper introduces U-SafeBench, which is a new benchmark for assessing the user-specific safety of LLMs. They then evaluate 20 popular LLMs on this benchmark and find that they perform very poorly, even if they had been specifically designed for safety. Finally, they introduce an inference-time (CoT-like) mitigation that breaks the response into two parts and demonstrate its effectiveness on LLMs out-of-the-box.

# Strengths

This work is an important and strongly motivated contribution to the literature of LLM safety and is a good fit for the Reliable ML Workshop.

The benchmark is extensive as it includes 130 different user profiles and 2700 real-world user instructions, and its development process (which includes human annotators) is generally well documented.

The experimental evaluation is comprehensive and it's interesting that basically every model (including the top performing proprietary ones) get a low score on the benchmark.

Finally, I found it interesting that the simple 2-step mitigation proposed in the paper already achieves such a good increase in the safety score.

# Weaknesses / Limitations

* One limitation of this paper is that the safety score is merely measured as the percentage harmful instructions in which the model refuses to respond. I think that there is more nuance in what would make an acceptable response in certain scenarios. For example, telling a pregnant woman where to buy alcohol is not that serious (considering that one can simply Google-search that), if it is also followed/preceded by a strong warning that it is not advisable. I would have liked to see more discussion of this choice and, ideally, a more nuanced construction of the dataset to take this into account.

* Using an 8b Llama model as LLM-as-a-Judge sounds a bit unreliable, especially considering that the paper shows that LLMs are bad at solving this task. It is noteworthy though that the judgements of this model generally align with those of human annotators. But I would like to see more discussion of why you think there is this very big discrepancy between producing and verifying answers on this problem. As a follow up to this: if this discrepancy is really that big, does it mean that we can get another effective mitigation strategy by using the LLMs in "verification mode" to judge their own answers before responding?

* I would like to see more evidence that capping the token limit to 100 tokens does not affect the results. For example, why couldn't it be the case that a model initially intends to fulfill the instruction, but then decides against it? This could be especially relevant in reasoning models that include some explicit thinking time, but I understand that it might be beyond the scope of the paper in its current form.

---

### Official Review · Reviewer_w6sf · 2025-09-20
**Build a dataset and find LLMs fail at adjusting answers to user profiles**

**Rating:** 5
**Confidence:** 3

**Review:**

Pros

New safety angle: reframes safety as user-specific instead of one-size-fits-all, with concrete motivating cases like alcohol purchase that flip from safe to unsafe depending on profile.

Detailed benchmark: 134 user profiles and 2700 instructions, across QA and agent tasks

Evaluates 20 LLMs and finds very low user-specific safety 14.1), much worse than general standards, which shows strong evidence of a real vulnerability

Cons

Mitigation is light: the CoT pipeline is a prompt tweak, a useful baseline, but not a principled system

Strong assumption not very practical. Assumes reliable access to user profiles. real systems face privacy, correctness, and consent issues.

Relies on LLM-as-a-Judge, which can inherit bias or drift from the judge model, missing human evaluation.